# Nexus between Building Information Modeling and Internet of Things in the Construction Industries

**Baydaa Hashim Mohammed** [1,2,*], **Hasimi Sallehuddin** [1], **Elaheh Yadegaridehkordi** [3], **Nurhizam Safie Mohd Satar** [1], **Afifuddin Husairi Bin Hussain** [4,5] **and Shaymaa Abdelghanymohamed** [6]

1    Faculty of Information Science and Technology, Universiti Kebangsaan Malaysia, Bangi 43600, Malaysia
2    Medical Instrumentation Department, AL-Esraa University College, Baghdad 10069, Iraq
3    Center for Software Technology and Management, Faculty of Information Science and Technology, Universiti Kebangsaan Malaysia, Bangi 43600, Malaysia
4    Pusat Citra Universiti, Universiti Kebangsaan Malaysia (UKM), Bangi 43600, Malaysia
5    Razak Faculty of Technology and Informatics, Universiti Teknologi Malaysia, Kuala Lumpur 54100, Malaysia
6    Department of Electrical Engineering, University of Technology, Baghdad 10066, Iraq
*    Correspondence: baydaa@esraa.edu.iq

**Abstract:** The process of integrating building information modeling (BIM) and Internet of Things (IoT)-based data sources is a recent development. As a generalization, BIM and IoT data provide complementary perspectives on the project that complement each other's constraints. Applying the concept of BIM-IoT in the construction industries which has been termed to have a high-risk factor could offer an improvement in the overall performance of the construction industries and thereby reduce the associated risks. This study aims to examine the potential of integrating BIM-IoTs in the construction industries by examining related published literature. Literature analysis revealed that the BIM and IoT have been extensively applied individually to several aspects of construction projects such as construction safety risk assessment, construction conflict management, building construction sustainability, and onsite construction process monitoring. However, there is scanty research awareness on the possibilities of BIM-IoT integration in the construction industries.

**Keywords:** building information modeling; Internet of Things; construction risk management; construction conflict management; construction innovation

## 1. Introduction

The entire construction process, starting with the location of the building and ending with its management and upkeep is encompassed in the concept of Construction 4.0 [1]. Governments in many countries are encouraging the building industry to embrace Industry 4.0 [2]. To enforce the electronic form of the construction process from project preparation, budgeting, building approvals, construction management, and building management and the application of these public construction contract principles, efforts are being made to consolidate dispersed investment and construction management powers [3]. When the industrial revolution occurred, people's feelings, thoughts, and perceptions of the world changed dramatically [4,5]. Everything from machines to humans to how they feel, think and see the environment could be altered as a result [6]. The construction industry has benefited from industry 4.0, which has brought digital technology such as sensor systems, intelligent machinery, and smart materials to the field of construction [7]. Using building information modeling (BIM), a project's digital information is centralized in one place, where it is utilized to maintain track of all the digital information about a project [8,9]. BIM is the appropriate starting point when it comes to the construction industry. To develop strong and innovative apps, it provides an additional layer of data that may interact and collaborate in real-time throughout the project life cycle [10]. Using

modern BIM technologies, the construction industry can be made more efficient and cost-effective [11]. With the application of open BIM, existing construction management systems can be integrated with BIM to enhance its power in the construction ecosystem [12]. Even though BIM has gained widespread acceptance in the construction industry and is worth the investment for many firms, few have taken full advantage of its benefits [13].

The Internet of Things (IoT) is a new internet breakthrough in which billions of smart things are linked together [14]. Data can be exchanged between several computers and digital equipment without the need for human intervention using unique identifiers. It is possible to provide network users with added value services like ensuring the privacy of shared data by integrating IoT sensors into internet-connected devices [15]. Using IoT in the construction industry has various benefits [16]. There are a variety of benefits associated with this approach, such as enhanced execution monitoring and control, higher quality, lower costs, and more time saved [17]. Since real-time data analytics are now widely available for usage in the context of rapid decision-making, it has also been expanded. Structure monitoring improves crisis management and emergency response capacities as well. The IoT can help with environmental issues like trash management, pond pollution, and flood concentration analysis. The introduction of new technology brings with it a variety of challenges that need to be overcome [18]. Method of introduction, public acceptance, and lack of information and expertise are all potential roadblocks. construction projects are relatively difficult with a high risk of failure that comes along with them severely restricts the potential use of new technologies. Despite these difficulties, the IoT has been implemented in the construction industry, with one of the most prominent applications being the monitoring and control of project execution in a variety of projects, including bridges, railways, tunnels, and onshore and offshore facilities, among others.

A few studies have concentrated on the integration of BIM and IoT to improve innovation in various areas of endeavours. Tang et al. [19] reported the integration of BIM and IoT devices in the Architecture, Engineering, and Construction (AEC) sector from the standpoint of domain application and integration approaches, as well as by shedding light on the present limits and significant topics for future research and development. Malagnino et al. [20] presented an extensive review on integrating BIM and IoT to create more energy-efficient and environmentally friendly environments. The authors inferred that, with the integration of BIM and IoT, the built environment can be better managed, and the environmental effect of the construction industry can be reduced. Lokshina et al. [21] investigated BIM, IoT, and blockchain technologies in the system design of a smart building. The authors viewed these understudies' technologies as complementing innovations that may work together to provide the safe storage and management of building-related data and information and to improve the IoT services offered.

As the integration of BIM and IoT is still in its early stages, it is vital to get an awareness of the current situation related to its implementation in the construction industry. Vital questions such as what are the most often encountered BIM and IoT device integration scenarios? What classification system should be used to classify these application domains? what is the best way to combine BIM with IoT devices? what are the constraints, both in terms of application domains and integration methodologies, when it comes to both? What research gaps that need to be filled, and where should we go from here to conduct more successful investigations? Furthermore, this study highlights the obstacles connected with implementing the integration of BIM and IoT in construction projects, as well as determining the most critical challenges associated with the implementation. To address these research questions, this study performed a comprehensive review of BIM and IoT implementation in the construction industry, including a summary of application areas, and integration methods used in existing studies. The study also examined inherent limitations and anticipated areas for further research.

## 2. Methodology

The approach used for the review process is depicted in Figure 1. The first stage entails the identification of the main purpose of the review and formulating appropriate questions that could serve as a guide. To achieve the set objectives and provide answers to the research questions, quality databases such as Scopus (www.scopus.com (accessed on 2 May 2022)) and Science Direct (www.sciencedirect.com (accessed on 2 May 2022)) were employed to search related published articles. This section details the date range, information sources, eligibility restrictions, and data coding technique. Keywords such as "Building information modeling", "Internet of Thing", "Building information modeling and Internet of Thing Integration", and "Construction industry" were employed to search the databases. The exclusion criteria include those articles that are not indexed in Scopus and Science Citation Index Expanded as well as articles published in the last ten years (2012–2022). The articles obtained from the search were quickly screened using the title and abstract to identify their relevance. The selected articles were carefully reviewed and analyzed to meet the study objectives and to answer the various questions.

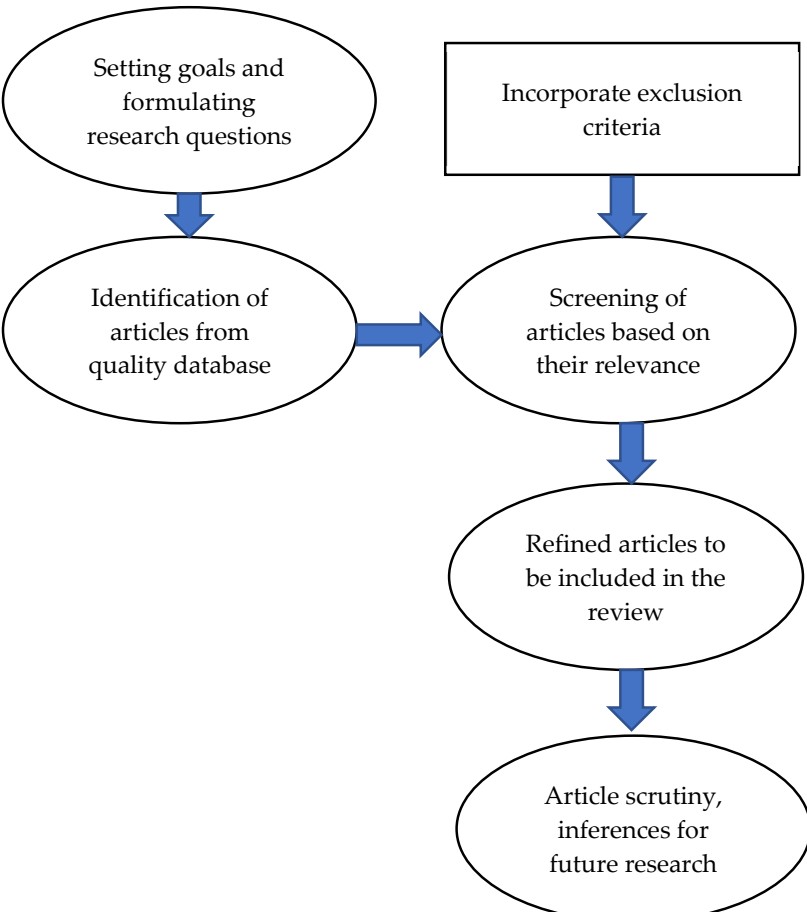

**Figure 1.** Methodology adopted for the review process.

## 3. Results and Discussion

### 3.1. Research Trends in BIM and IoTs in Construction Industries

Scopus database was employed to search for published articles related to BIM and IoT adoption in the construction industries as well as the BIM-IoT integration in the construction industries around the world. Keywords such as BIM in the construction industry, IoT in the construction industry, integrating BIM and IoT in the construction industry and so on were employed for the search. As shown in Figure 2a, the analysis of the search results shows that there has been a rise in research interest in BIM and IoT adoptions

in construction between 2012 and 2022. However, there is higher research interest in BIM adoption compared to IoT adoption in construction. Perhaps, this could be attributed to the fact that IoT applications in the construction industries in a nascent stage and is gradually gaining research interest. Figure 2b shows that research interest in BIM adoption in the construction industries is prevalent among researchers from the United States, the United Kingdom, and China. Also, researchers from countries such as Australia, Malaysia, South Korea, Hong Kong, Canada, and Spain are showing increasing interest in BIM adoption in the construction industries. Apart from China, there are low research interests in IoT adoptions in construction industries from other countries. Compared to individual BIM and IoT adoptions in construction industries, the research interest in the integration of BIM and IoT in construction industries is lower as shown in Figure 3a. The research interest in the integration of BIM-IoT applications in the construction industries is predominant in China, Hong Kong, the United Kingdom, the United States, Australia, Singapore, Canada, the Netherlands, and Nigeria. This trend is consistent with the studies of Bui et al. [22] According to the findings of construction industries in developing nations have several challenges due to the socioeconomic and technological context in which they operate. A dearth of IT-literate individuals, as well as the absence of national BIM implementation plans, are just a few of the obstacles that are hindering widespread BIM adoption. The authors inferred that construction industries in developing nations rely on subcontracting information technology services or contracting out software development. Although BIM is becoming increasingly popular in developed nations, applications in developing countries are still rare [23]. Olawumi & Chan [24] suggested some benchmarking models for BIM implementation in developing countries. The concepts developed by the authors include innovative strategies at the BIM process level, innovative strategies at the BIM product level, and measures of good practices. These ideas were discovered to contain blueprints that can enhance BIM products and processes, as well as related technology, to make adoption and deployment simpler and thus more secure in developing nations.

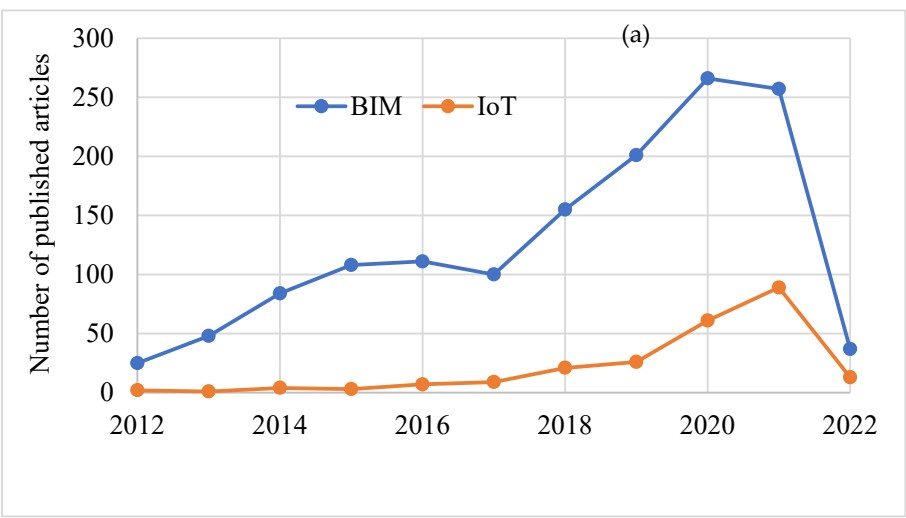

**Figure 2.** *Cont.*

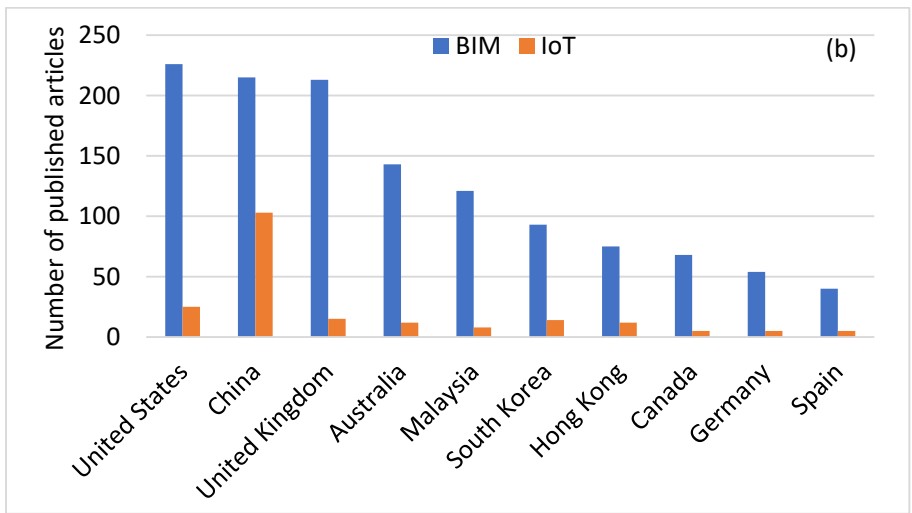

**Figure 2.** (**a**) Trend of articles published in BIM and IoTs adoption in construction industries; (**b**) Countries of Affiliation of the research (Data obtained from Scopus Database).

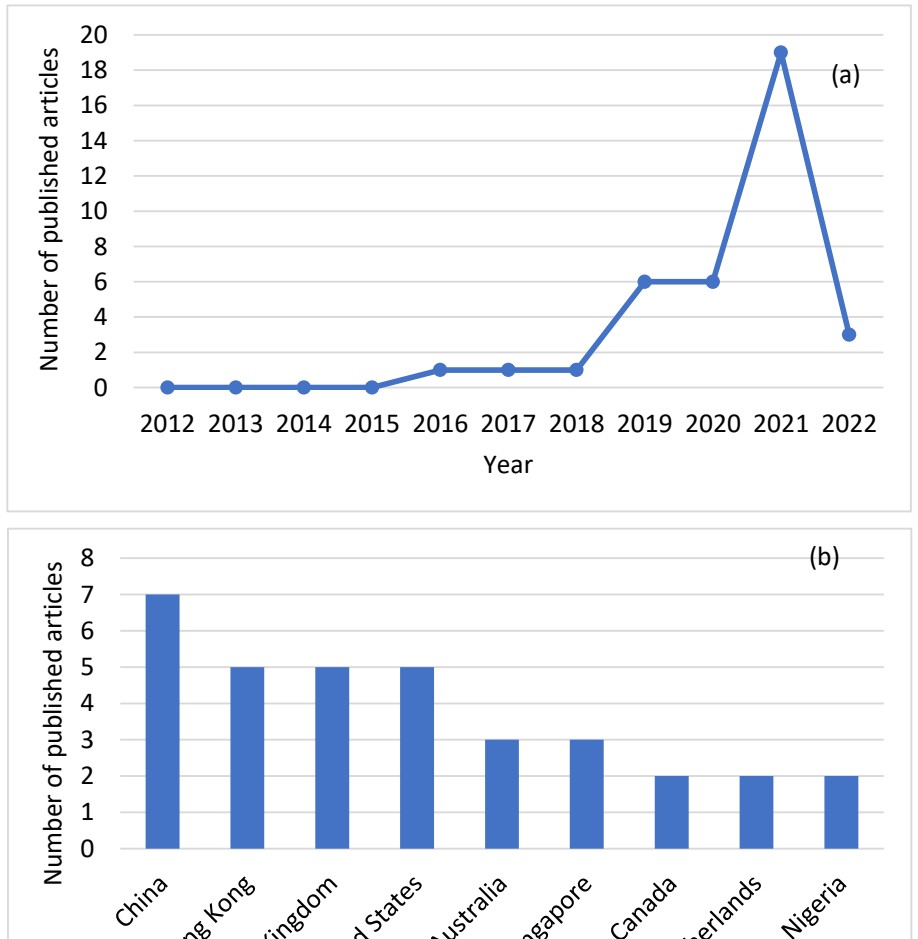

**Figure 3.** (**a**) Trend of articles published in BIM-IoTs integration in construction industries; (**b**) Distribution of BIM-IoT studies according to countries. (Data obtained from Scopus Database).

### 3.2. An Overview of BIM in the Construction Industry

In recent years, digitalization has had a significant impact on several industrial sectors, enhancing productivity, product quality, and product diversification. Digital technologies are gradually being used in the AEC industry to design, build, and operate buildings as well as other construction assets [25]. Other industries, on the other hand, use digital data throughout the entire process chain. Physical plots on paper or in a limited digital format are still used to convey most information. When a completed facility is being designed, built, and operated, as well as during the crucial handovers between these phases, such interruptions to the information flow are common. The creation and execution of the physical infrastructure in the construction industry involve a wide range of parties with diverse backgrounds and areas of competence. To ensure a successful construction project, these stakeholders must regularly interact and reconcile. Most projects now include the delivery of technical drawings in the form of cross-sections, elevations, and detail drawings (both horizontally and vertically). Using software that resembles centuries-old sketching skills, these images were created. Computers, on the other hand, are unable to properly comprehend line drawings. Only a portion of the data can be fully analyzed using statistical methods. Thus, the enormous potential of information technology for project management and building operations is not fully realized. The inconsistency of different technical drawings is a significant problem. It is a major problem because the drawings are generated by professionals from a variety of design disciplines and companies. Due to the rapid occurrence of inconsistencies, which typically go unnoticed until the construction project is complete, additional costs can be incurred for ad-hoc remediation on the construction site. Traditionally, design changes are denoted in drawings by revision clouds. To perform any downstream analysis, calculation, or simulation, it is necessary to manually enter data from technical drawings, which increases both the burden and the risk of human error. A similar scenario occurs after the structure is completed and the owner receives the information. It takes a lot of time and effort to extract building data from drawings and paperwork into a facility management system. Previously available digital data has been erased and must be reconstructed manually at each of these information exchange locations. Employing BIM in the construction industries allows for far more extensive use of computer technology in the design, engineering, building, and operation of built facilities. BIM is used instead of traditional drawings to store, preserve, and communicate data during the design and construction process. Coordination of design activities, simulation integration, construction process setup, and control, as well as the transfer of building information to the operator, are greatly improved by this approach. In construction projects, arduous and error-prone work is avoided, resulting in increased productivity and higher quality, by minimizing the amount of human data entry and enabling the re-use of digital information.

Several studies have been reported on BIM applications in the construction industries as summarized in Table 1. Studies have shown that the construction industry has a high prevalence of occupational fatalities and injuries due to the dynamic nature of the working environment. In line with this Lu et al. [26] offered a new quantitative construction safety risk assessment method for building projects in the design phase using an integrated BIM. The authors' findings revealed that a plug-in that connects BIM and safety risk data to automatically evaluate construction safety risk was developed thereby allowing architects and structural designers to make faster design choices. The role of BIM in construction conflict management has been investigated by Sacks et al. [27]. The study utilized a questionnaire survey method to identify the most prevalent factors that influence BIM implementation in the construction industry. They revealed that time and expense constraints, as well as poor construction project management scheduling and update requirements, and the late release of design information or drawings, were the most significant contributors to many of the problems encountered in the implementation of BIM. Similar to the work of Sacks et al. [27], Ozturk [28] reported the implementation of BIM in different aspects of construction projects. Based on the authors' analysis, data sharing issues, integration inefficiencies, lack of collaborative design, construction operation, facility management,

and communication obstacles in project and construction management were observed to be the dominant hindrances to BIM adoption in the construction industry. BIM has also been reported to be vital in construction conflict management as reported by Charehzehi et al. [29]. The analysis of the questionnaire survey revealed that insufficient monitoring of construction projects, scheduling and updates requirements, failures to understand and correctly bid or price of the works, delays in running bill payment, inadequate contractors' management, supervision, and coordination, errors and omissions in design that originate from time, cost, quality, and documentation were the critical conflict factors in Malaysia's construction industry. The incorporation of BIM was observed to be capable of resolving the identified construction conflicts. The effect of implementing BIM in the Turkish and Chinese construction industries has been investigated by Aladag et al. [30] and Xu et al. [31], respectively. Using a qualitative research method, Aladag et al. [30] revealed that the acquisition of firms is the most important driving factor for the Turkish construction industry to adopt BIM, while "organizational structure and culture" is the most significant hurdle. Xu et al. [31] on the other hand employed a semi-structured interview and questionnaire survey to identify the most important criteria to adopt BIM in the Chinese construction industry. The analysis revealed that he most important criteria connected to industrial requirements in order to adopt BIM in Chinese construction industry are customer demand/contract responsibilities and need for collaboration, coordination, communication, and interoperability across stakeholders. Besides being used for conflict management, BIM has been reported to be applicable as an asset management tool in the construction industry. Based on the conceptual development of Guillen et al. [32], the advantages of using BIM for asset management were not clearly defined. The methodologies and application frameworks have to be proposed and evaluated to generate reference use cases that will allow extending the knowledge about asset management enabled by the BIM model as the platform for an asset management information system. Furthermore, the application of BIM in building design optimization for sustainability using particle swarm optimization modeling has been reported by Liu et al. [33]. The study revealed that the particle swarm optimization method can expand the search space for optimal design solutions and minimize the processing time for optimal design outcomes, which is extremely beneficial to designers in delivering an environmentally and economically friendly design scheme. BIM has been found helpful in the onsite construction process controlling the liquefied natural gas industry. Using three-dimensional modeling, Wang et al. [34] found that extending the BIM solution to the site could help to handle more real-world issues such as low productivity in collecting information, a proclivity for making mistakes in assembly, and low communication as well as problem-solving efficiency. The effectiveness of BIM implementation during different construction phase has been reported by Fan et al. [35]. From the pilot survey and qualitative method, the authors inferred that the increased usage of BIM by contractors, the amount of money that can be charged as a premium will start to decline.

**Table 1.** Summary of Literature on BIM implementation in the construction industry.

| Index | Area of Application | Methodology | Findings | Reference |
|---|---|---|---|---|
| Scopus and SCIE | Construction Safety Risk Assessment | A theoretical framework and a developed plug-in | BIM and safety risk data are linked together in Autodesk Revit, and a plug-in is made to help architects and structural designers quickly choose design alternatives. | [26] |
| Scopus | Construction conflict management | Questionnaire surveys, Analytical hierarchy process, and multi-attribute utility technique | The study revealed that time and expense constraints, as well as poor construction project management scheduling and update requirements, and the late release of design information or drawings, were the most significant contributors to many of the problems encountered in the implementation of BIM | [27] |
| Scopus and SCIE | Architecture, engineering, construction, operation, and facility management | Bibliometric analysis | Data sharing issues, integration inefficiencies, lack of collaborative design, construction, operation, and facility management, and communication obstacles in project and construction management continue to hinder BIM adoption. | [28] |
| Scopus | The use of BIM in construction conflict management. | Questionnaire survey | A large number of the important reasons that cause construction conflict management stem from issues related to the project's schedule and cost, quality, and documentation. | [29] |
| Scopus | Turkish Construction Industry | Qualitative research methods | Acquisition of firms" was the most important driving factor for the Turkish construction industry to adopt BIM, while "organizational structure and culture" is the most significant hurdle. | [30] |
| Scopus and SCIE | Chinese construction industry | Semi-structured interview and Questionnaire Survey | The most important criteria connected to industrial requirements to adopt BIM are customer demand/contract responsibilities and the need for collaboration, coordination, communication, and interoperability across stakeholders. | [31] |
| Scopus | The application of BIM as an asset management tool in the construction industry. | Conceptual development | The advantages of BIM for asset management have yet to be clearly defined. Methodologies and application frameworks have to be proposed and evaluated to generate reference use cases that will allow extending the knowledge about AM enabled by the BIM model as the platform for an asset management information system. | [32] |
| Scopus and SCIE | BIM application in building design optimization for sustainability | Particle Swam Optimization modeling | The study revealed that the particle swarm optimization method can expand the search space for optimal design solutions and minimize the processing time for optimal design outcomes, which is extremely beneficial to designers in delivering an environmentally and economically friendly design scheme. | [33] |

**Table 1.** *Cont.*

| Index | Area of Application | Methodology | Findings | Reference |
|---|---|---|---|---|
| Scopus | The effect of BIM during the constructionphase | Pilot Survey and qualitative method by Interview. | With the increased usage of BIM by contractors, the amount of money that can be charged as a premium will start to decline. | [35] |
| Scopus and SCIE | Onsite construction process controlling for liquefied natural gas industry. | Three-dimensional modeling | It is shown that by extending the BIM solution to the site via AR's "hand," it may handle more real-world issues such as low productivity in collecting information, a proclivity for making mistakes in assembly, and low communication and problem-solving efficiency. | [36] |

*3.3. An Overview of the Role of IoTs in Construction Industries*

The application of IoT in the construction industry has several benefits (Table 2). Some of the benefits include better project execution monitoring, effective control, improved quality, cost savings, and time savings [37]. Due to the availability of real-time data analytics, it has also been broadened to be utilized for quick decision-making. By establishing effective monitoring of the structure, IoT also enhances crisis management and emergency response [38]. In addition, IoT has been utilized to monitor and manage project executions in a variety of construction projects [39]. In terms of productivity, the construction industry is the least digitalized sector [40]. The incorporation of IoT in the construction industries has been termed Smart construction [41]. "Smart construction" refers to the use of cutting-edge technology to improve productivity and decision-making in the construction sector, and this technique is referred to as "smart construction" since it makes use of the newest technologies [42]. With the use of digital technology, smart construction aims to revolutionize the construction industry by improving the efficiency and effectiveness of construction resources including machinery, gadgets, components, and labourers [43]. As a branch of intelligent technology, intelligent construction is an important part of it [44]. Many aspects of the construction business have been transformed by technological advances such as automation, cyber-physical systems, cloud computing, cognitive computing, and immersive technologies. IoT, a new concept in digital technology that is quickly gaining acceptance, is continuously extending the range of applications for smart buildings [45]. These researchers want to see if the IoT has any impact on how well smart construction is delivered. Modern construction approaches including offsite building, lean construction, smart assembly, and so on are examined in this paper [46]. From the standpoint of current building practices, such as offsite construction, lean manufacturing, and smart assembly, it also highlights upcoming IoT application themes [47]. On-site and off-site manufacturing, assembly and logistics in real time, health and safety, information and communication management, energy management, waste minimization, remote inspection, and lean construction management are all topics covered in the paper [48]. Based on the existing literature, there is a better grasp of how digital technologies may be utilized to transform the construction process and a strong foundation for future research into the IoT and smart building. Construction companies have a chance to offer cutting-edge solutions to the problems raised by the numerous IoT applications.

**Table 2.** Summary of Literature on IoT implementation in the construction industry.

| Index | Area of Application | Methodology | Findings | Reference |
|---|---|---|---|---|
| Scopus and SCIE | Application of Blockchain and IoT in the construction industry | Conceptual analysis | Real-world IoT applications in monitoring construction site health and safety, assessing the functioning of structural parts like bridges, and managing facilities was reported. The authors proposed using IoT in larger contest in the construction industry. | [14] |
| Scopus and SCIE | Trend of IoTs in construction industries | Conceptual and Literature analysis | The following were identified as key drivers of IoT adoption in the construction industry: interoperability; data privacy and security; adaptable governance frameworks; and adequate business planning and modelling. | [17] |
| Scopus | Solar photovoltaic power generation technology and building construction based on the Internet of Things | Real-life design and Simulation | IoT and ZigBee wireless sensor network were effective to study the distributed solarenergy devices incorporated to building construction project. The joint design of solar energy devices and buildings are of great significance to the development of photovoltaic construction industry. | [49] |
| Scopus | IoTs in Malaysia Construction industries | Questionnaire Survey | The findings suggest that among the numerous types of IoT applications utilized by construction industry participants include social media platforms like WhatsApp, Telegram, and Facebook for discussion and communication, email for information and communication exchange, and website use as a source of reference to gather data on corporate profiles, activities, and policies, among others. quotes on price | [50] |
| Scopus | Role of IoTs in Developing energy efficient organic bricks in the Construction industries | Experimental Design | The study demonstrated that eco-friendly bricks made from mine waste had decreased thermal conductivity, high strength, and are light in weight. | [51] |
| Scopus | IoTs adoption in Constructions industries located in East Malaysia | Questionnaire Survey | Increasing IoT acceptance among contractors in the construction industry, and the Malaysian government may take steps to accelerate IoT adoption in order to embrace the Industrial Revolution 4.0 (IR 4.0) in the country | [52] |
| Scopus and SCIE | Improved Lean Construction management using IoTs | Design Science Method | Based on the results, the authors created a communication framework that automates system–system, system–human, and human–system communication activities across the supply chain and project lifecycle. | [53] |
| Scopus and SCIE | IoTs in construction industry revolution 4.0 | Questionnaire survey | Lack of safety and security, lack of defined standards, lack of benefit knowledge, poor IOT implementation, and lack of resilience in connectivity are the most prevalent difficulties in the study. | [54] |

**Table 2.** *Cont.*

| Index | Area of Application | Methodology | Findings | Reference |
|---|---|---|---|---|
| Scopus | IoT enabled Prefabricated construction | Experimental Design and Fabrication | A multi-dimensional IoT-enabled Building Information Modeling (BIM) platform (MITBIMP) for achieving real-time tracking in prefabricated building was achieved | [55] |
| Scopus | IoTs in Nigeria Construction Industries | Questionnaire Survey | The results indicate that power availability, cost, limited customer demand, and issues with online security and data volume are the most significant barriers to the adoption of the Internet of Things in the construction business. | [56] |

The technological potentials and obstacles to combining IoT with Blockchain in the construction industries have been examined by Elghaish et al. [14]. The concept dealt with a real-world analysis of blockchain and IoT in the construction industries based on the analysis of various studies. The study revealed that real-world IoT applications in monitoring construction site health and safety, assessing the functioning of structural parts like bridges, and managing facilities are feasible and enhance project performance in the construction industry. The authors proposed using IoT in a larger context in the construction industry. The application of IoT in solar photovoltaic power generation and building construction projects has been reported by Wu et al. [49]. The study revealed that IoT and ZigBee wireless sensor networks were effective to study the distributed solar energy devices incorporated into building construction projects. The joint design of solar energy devices and buildings is of great significance to the development of the photovoltaic construction industry. The effect of IoT implementation in Malaysian construction industries has been reported by Mahmud et al. [50]. The study which employs a questionnaire survey method revealed that social media platforms like WhatsApp, Telegram, and Facebook for discussion and communication, email for information and communication exchange, and websites are used as a source of reference to gather data on corporate profiles, activities, and policies, among others. quotes on price were among the numerous types of IoT applications utilized by construction industry participant [51] investigated the role of IoTs in developing energy-efficient organic bricks in the construction industries When compared to regular bricks, the heat transmitted from the outside to the interior of the walls of the model room created with IoT-perlite bricks was at least 2 °C lower. Lower thermal conductivity leads to energy savings, and research showed that IOT-perlite bricks saved 8% of energy. The study demonstrated that eco-friendly bricks made from mine waste had decreased thermal conductivity, high strength, and were light in weight [52] investigated the adoption of IoTs among contractors in East Coast Malaysia construction industries. The questionnaire survey analysis revealed that attitudes, awareness, preparedness, and impediments are all factors influencing changes in IoT adoption among contractors. The construction industry's significant association with IoT adoption was observed to have been expanded appropriately. The study also provided some important recommendations for increasing IoT acceptance among contractors in the construction industry. Dave et al. [53] investigated improved lean construction management using IoTs. Based on the analysis of the results, the authors developed a communication framework that allows them to automate different communication tasks completely or partially across the supply chain and the lifespan of building construction projects by the leveraging system–system, system–human, and human–system communication. Gamil et al. [54] reported IoTs in the construction industry revolution 4.0. The findings of this study show that the lack of safety and security, lack of defined standards, lack of benefits knowledge, poor IoT implementation, and lack of resilience in connectivity are the most prevalent difficulties in this study. To find out if construction workers are aware of the benefits IoT may bring to construction projects, this

study also investigated their knowledge of IoT and whether or not it can be implemented and expanded in such projects. Ghosh et al. [17] analyzed the trend on IoTs in construction industry. Based on the analysis, the primary implications of IoT adoption in the construction sector have been highlighted as high-speed reporting, total process control, data explosion resulting in deep data analytics, and severe ethical and regulatory requirements. The following were identified as key drivers of IoT adoption: interoperability; data privacy and security; adaptable governance frameworks; and adequate business planning and modelling. IoT enables prefabricated construction has been investigated by [55]. The authors offered a multi-dimensional IoT-BIM platform for achieving real-time tracking in prefabricated building construction. For the purpose of designing the IoT-BIM platform, design considerations for an RFID Gateway Operating System, visibility and traceability tools, data source interoperability services, and decision support services were provided.

### 3.4. Nexus between Building Information Modeling and Internet of Things

As a result of advancements in technology, building systems have gotten more complicated than ever before to account for things like advanced technologies, security concerns, and eco-friendliness. Hence. Integrating BIM with IoT will help to overcome these challenges and helps to proffer lasting solutions to the concerns.

Building Information Modeling (BIM) combined with real-time data from IoT devices creates a strong concept for improving construction and operating efficiency [57]. Integrating real-time data streams from the constantly growing number of IoT sensing devices to strong BIM models opens up a world of possibilities. Nevertheless, considering BIM and IoT device integration research is still in its infancy, it is necessary to comprehend the current state of BIM and IoT device integration. BIM and IoT data provide unique views on the construction project that balance each other's limitations. At the construction component level, BIM models provide high-resolution representations of the project. Whereas data collection for a construction project can be enhanced using an IoT platform that provides real-time and recordable status from actual construction and activities. An extensive study by Tang et al. [19] shows that BIM-IoT integration in the construction industries is applicable in four different domains namely: Construction operation and monitoring; health and safety management; construction logistics and management; facility management. The details of the BIM-IoT integration possibilities are summarized in Figure 4. The specific applications of the BIM-IoT integration in the construction operation and monitory domain include onsite environment monitory, resource monitory, communication, and collaboration as well as construction performance and progress monitory. In the health and safety management domain, the specific applications are health and safety training as well as onsite monitoring for health and safety. The construction logistic and management domain has automation in fabrication and lean construction as possible areas of BIM-IoT integration. While the facility management domain has building operation and maintenance, building performance management, energy management as well as disaster and emergency response as possible areas of BIM-IoTs integration opportunities.

Various ways to facilitate the integration of IoT and BIM systems have previously been presented in the literature. Wan & Bai [58] presented the integration of BIM and IoT in construction logistic management. The study evaluated the new characteristics of building logistics management in the context of big data, developed a collaborative logistics management system with data-driven and BIM technology and assessed the collaborative logistics management solution based on big data. Malagnino et al. [20] opined that integrating BIM and IoT could produce a smart and sustainable environment. The authors suggested a modular design for integrated BIM-IoT systems based on their analysis. The authors include several blocks for the insertion of data from BIM systems and IoT middleware. All data were collected into a Central Control Unit, which is composed of several highly specialized sub-blocks that facilitate data collecting, storage, sharing among the unit's other blocks, presentation, and the application of Artificial Intelligence algorithms to fine-tune acquired data. Liang & Liu [59] investigate the integration of BIM and IoT in the

construction safety risk of underground engineering construction. The authors analyzed the new characteristics of BIM and logistics management in the era of big data and IoT, building the logistics collaborative management platform based on big data and BIM technology, as well as an evaluation of the logistics collaborative management platform.

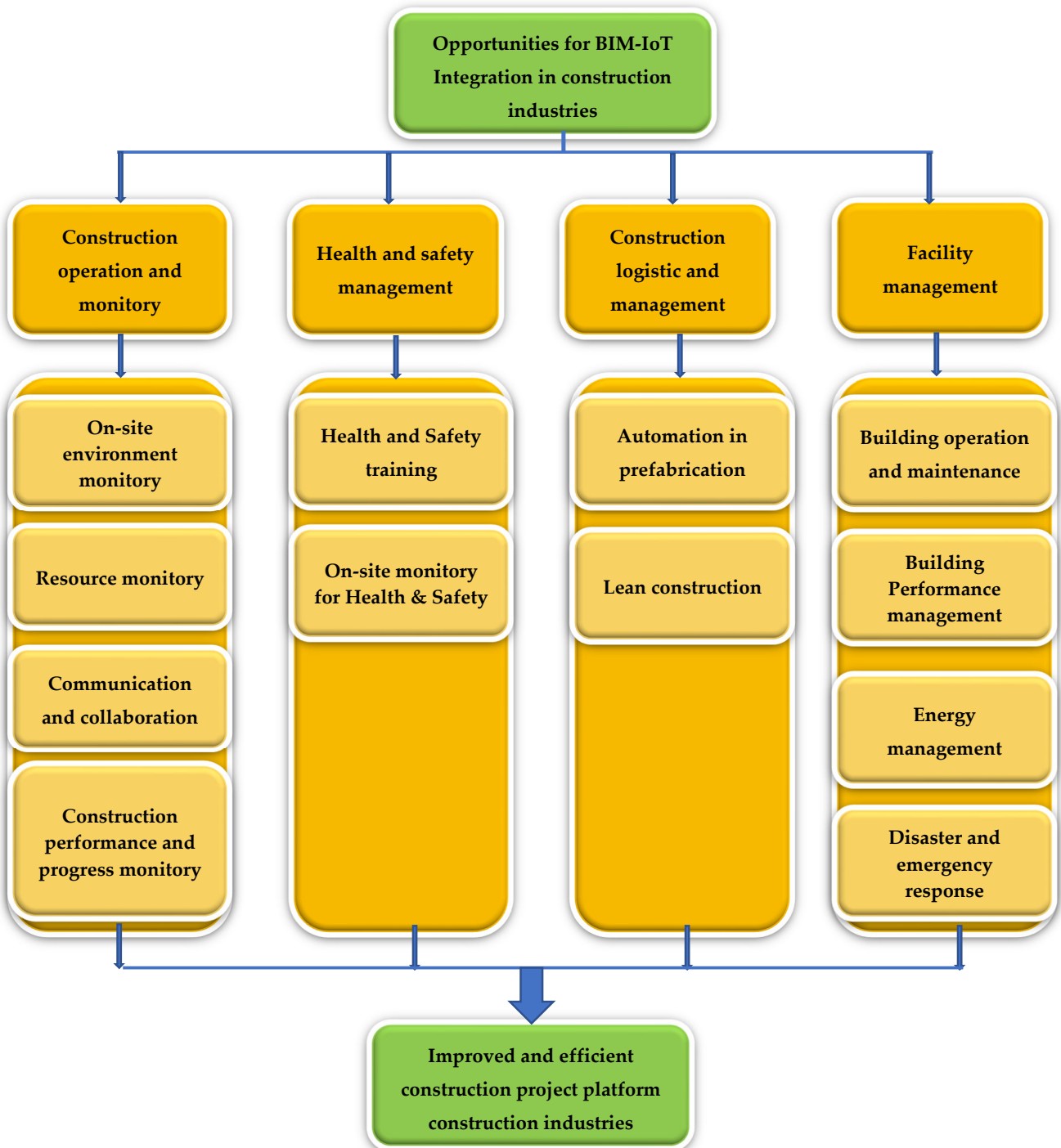

**Figure 4.** Opportunities for BIM-IoT integration in the construction industries.

*3.5. Geographical Distribution and Major Themes of BIM and IoT Studies*

Studies on technological adoption of BIM and IoT have been widely investigated in different countries as shown in Figure 5. There is significant awareness of BIM and IoT adoption in Malaysia as indicated by the number of research articles published. Rahim

et al. [60] investigated BIM awareness among Malaysian contractors. The objective of this study was to investigate the level of understanding that Malaysian contractors have regarding the role that BIM plays in achieving sustainability on all fronts, including the economic, environmental, and social fronts. A survey included 133 different contractors, ranging from grade G1 to G7. The findings showed that most respondents were aware of the contributions that BIM makes toward environmental sustainability, in addition to the contributions that BIM makes toward the other two pillars of sustainability, economic and social. Therefore, it is necessary to educate the stakeholders in the construction sector and give information that is based on reality as part of a process to generate a better understanding and wider exposure, and to convince them to apply BIM innovation. Othman et al. [61] in their study reported the level of BIM implementation in Malaysia. The purpose of this study was to examine the adoption of BIM by Malaysian business organizations. Based on the findings, only 13% of the 268 respondents in the public and private sectors reported utilizing BIM in their organizations, indicating that Malaysia is still far from where it should be in terms of BIM implementation, according to the study's findings. There was a lack of awareness, expenses, delayed adaptation, the lack of a clear guideline to assist organizations and policymakers toward BIM implementation, and the fact that BIM was not mandated in adequate time were identified to be the causes of the slow adoption. Roger et al. [62] investigated the adoption of BIM in Malaysia from the perspective of engineering consulting service firms. The findings demonstrate that the organizations have a BIM concept that is consistent with industry standards; nonetheless, the primary impediments to implementation are a lack of well-trained employees, advice, and government backing. Nonetheless, the enterprises were ready to embrace BIM within two years, citing market needs and competitive advantage as the key factors. The adoption of IoT in the Malaysian construction industry has been reported by Ibrahim et al. [63]. The findings revealed that IoT adoption in the Malaysian construction sector is growing but still lags behind other Asian countries. The authors further revealed that the adoption of IoT will have a bright future with encouragement from the Malaysian government and backing from the Department of Public Works and the Construction Industry Development Board since both the private and public sectors are aware of the benefits of doing so. Apart from Malaysia, BIM and IoT adoption have been investigated in other countries such as Vietnam, the United Kingdom, United Arab Emirate, Taiwan, South Africa, Singapore, Saudi Arabia, Palestine, Nigeria, New Zealand, South Korea, Kenya, Jordan, Iran, India, Hong Kong, Ghana, Germany, and Finland. The main theme of the various studies centered on BIM integration, the barrier to IoT adoption, BIM adoptions, BIM contributions, BIM effectiveness, BIM uptakes, hindrances to BIM implementation, IoT adoption, and evaluation of BIM. Among the various themes, BIM and IoT adoption has been widely investigated.

Figure 6 shows that the major themes of the studies of BIM and IoT in the construction industry are diverse. However, the barriers and adoption of the implementation of BIM and IoT in the construction industries are the most pronounced. Other themes such as factors affecting BIM and IoT, augmented reality and BIM integration, and level of acceptance is sparingly investigated.

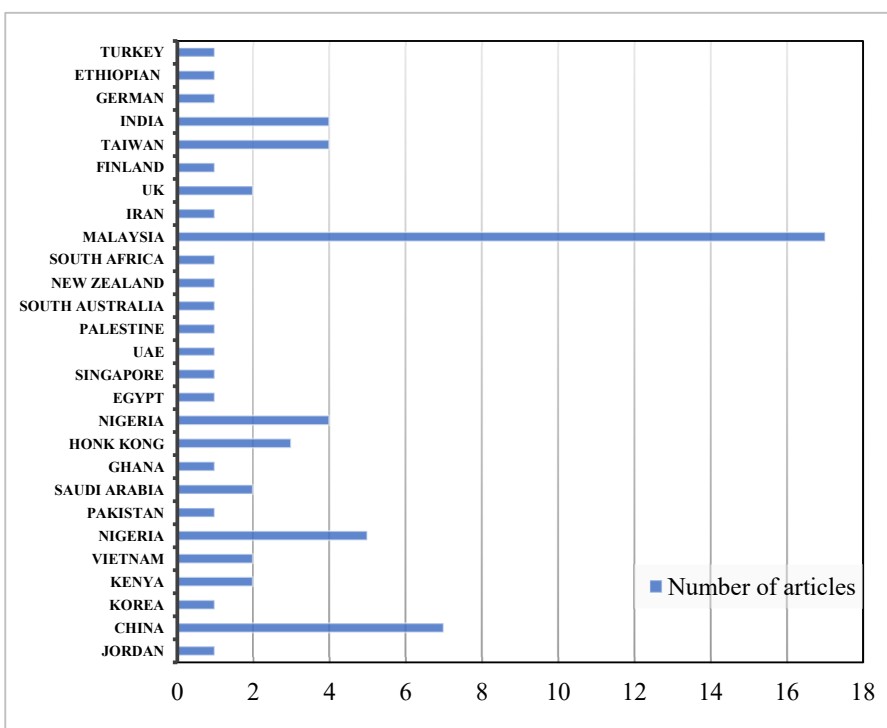

**Figure 5.** Geographical distribution of BIM and IoT studies.

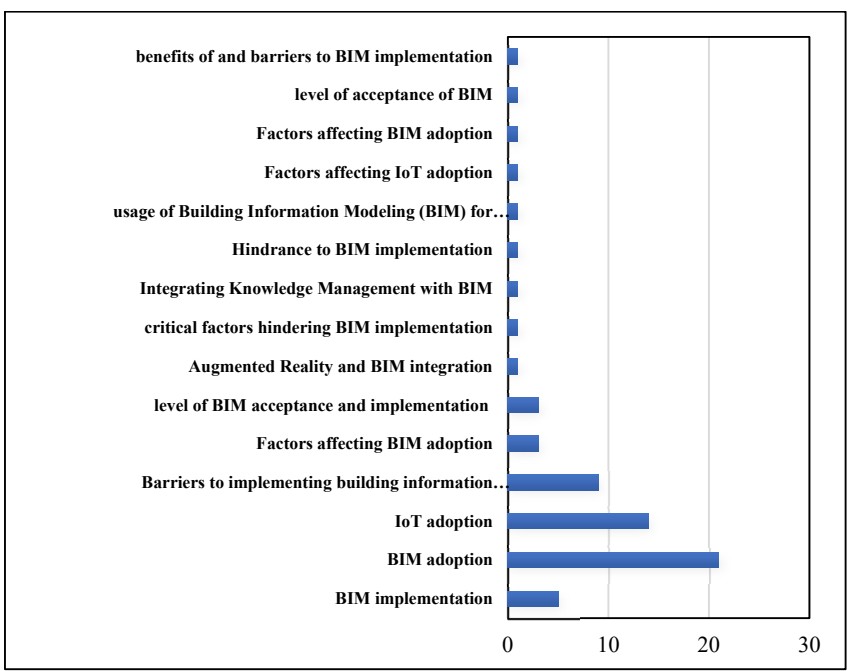

**Figure 6.** Major themes of the studies on BIM and IoT.

*3.6. Theories Employed for BIM and IoT Adoptions*

As it can be seen in Figure 7 that the TAM, TOE, IDT, HOT-Fit, and Institutional theories have been employed to gauge the amount of BIM and IoT acceptance by both people and organizations. TAM which was propounded in 1989 by Davis is one of the most extensively used acceptance theories. It discusses the importance of attitude, intention, and action in embracing or rejecting innovations. External factors, according to this model, impact Perceived Ease of Use, Perceived Usefulness, and attitude. Attitude results in behavioural intent. Using the Technology Acceptability Model, Acquah et al. [64] examine

the extent of BIM acceptance in Ghana's construction sector. Hypotheses were constructed, a questionnaire was designed, and a survey was performed among construction industry experts based on the TAM constructs. Chen et al. [65] employed the TAM as a theoretical basis for investigating the willingness for IoT adoption in Taiwan's construction industry.

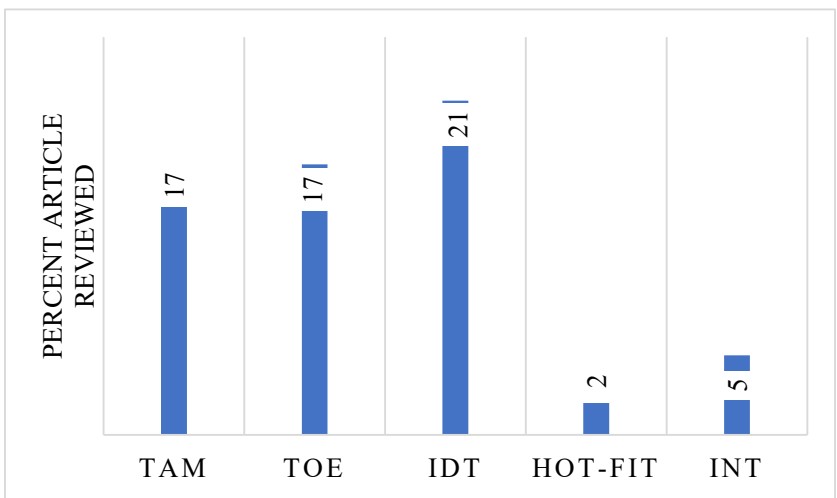

**Figure 7.** Overview of the theories adopted for BIM and IoT.

Similarly, the IDT which was popularized by Rogers and York in 1995 is also one of the most widely used acceptance theories. The IDT theory's construct incorporates observability, complexity, compatibility, trialability, and relative benefit. The IDT theory was adopted by Saka et al. [66] as a foundation to investigate the drivers of sustainable adoption of BIM in Nigerian construction small and medium-sized enterprises. According to the findings, organizational preparedness is of the highest significance for the proliferation of BIM in SMEs. Also, the independent drivers, which are the most significant, comprise BIM features, and internal and external environment drivers, and therefore represent the BIM adoption as a complex socio-technical system. In a similar study, Le et al. [67] also employed the IDT as a basis for investigating BIM implementation in the Vietnamese construction industry. The findings revealed that the BIM team functions as a tool that facilitates BIM implementation; nonetheless, there was a contradiction between the duties of the BIM team and the organization.

Besides, the TAM and IDT theories, the TOE has also enjoyed some level of popularity. The TOE can be defined as the enterprise-level innovation process. The TOE frameworks classify characteristics into three categories. The first dimension is technological, the second is organizational, and the third is environmental. Relying on the TOE theory, Saka and Chan investigated the profound barriers to BIM adoption in the construction of small and medium-sized enterprises. The findings indicated that the barriers to BIM adoption are socio-technical and that SMEs have the desire to accelerate BIM adoption by concentrating more on their internal environment. Chen and Yin [68] establish a study model that incorporates the important success elements linked to BIM technology, the construction firm, and the environment in the Chinese construction sector, based on the TOE theory. The authors discovered that BIM's relative benefit was a key driver in its acceptance, whereas its complexity was a deterrent. Furthermore, management support was a crucial factor in BIM adoption. Organizational preparedness, on the other hand, was important for engineering consulting businesses but not for construction companies. Surprisingly, no persistent substantial influences of any environmental variables were found by the authors.

The institutional theory which was propounded by Scott et al. [69] has been used for BIM and IoT. The theory focuses on the function that the institutional environment plays in creating behavioural changes and attaining social legitimacy. Isomorphisms are the foundational building blocks of this theoretical framework. Institutional theory:

contributing to a theoretical research program The study of changes that occur because of pressure exerted by an outside entity is known as coercive isomorphism. The goal of mimetic isomorphism is to replicate the hierarchical structure of an existing organization in the expectation of achieving the same levels of success as other organizations. The phenomenon that is known as normative isomorphism refers to the pressure that comes from regulatory authorities and practitioners interested in licenses and certificates. Based on institutional theory, Osman et al. [70] investigated BIM adoption for quantity surveying firms. The authors opined that by boosting the effectiveness of BIM adoption, organizations require direction and proper techniques. Quantity surveying organizations will be better able to commit appropriate resources to reach their objective if they grasp the major criteria for BIM adoption. Similarly, Institutional theory served as a basis for investigating the barriers to BIM adoptions in SMEs as reported by Saka and Chan [71].

The least studied theory based on the literature is the HOT-fit. The HOT-fit theory was initially introduced by Swedish academics in the 1980s to enhance the level of safety in the nuclear power sector. The HOT-fit idea draws a line of differentiation between individuals and the organization. Humans are rigorously considered as individuals due to the fact that their relevance is based on their abilities, expertise, experiences, and existing relationships with other people, all of which are essential to complete a job or altering a business process. This 'human' factor considers not just an individual's cognitive, psychological, and social traits, but also their biological and cognitive makeup. An organization is a representation of the formally and informally ordered and structured way the task is carried out. Therefore, job descriptions, hierarchical positions, duties and powers, policies, company objectives and strategies, rules, procedures, cultural elements, and linkages between system components and subsystems are all considered to be a part of the 'organization' element. In terms of the 'technology' component, there is the possibility of categorizing technical systems as either main or secondary. Primary technical systems are those that pertain to manufacturing equipment, whilst secondary technical systems are those that pertain to administration and processes. Based on the HOT-fit theory, Papadonikolaki et al. [72], investigated an IoT-enabled platform for the production of housing in Hong Kong. This study allows significant stakeholders to have a better understanding of the external and internal circumstances of prefabrication development in Hong Kong.

*3.7. Critical Factors Influencing BIM and IoT Adoption and Implementation*

A wide range of various factors for BIM and IoT intention to adoption have been investigated in the literature. These factors of BIM and IoT cut across different sectors such as energy management, construction monitoring, health-and-safety management, and building management. BIM and IoT integration research, on the other hand, is still in its infancy, with most studies being theoretical and conceptual in scope.

Figure 6 summarizes the various factors extracted from the literature reviewed and subsequently used for the development of the hypotheses. BIM and IoT in construction industries are beneficial, and a framework for considerations during deployment and use has been developed. To begin, it was necessary to recognize these factors. Following the review of the prior literature, the factors have been extracted from the models and frameworks in implicit ways. As shown in Figure 6, Several studies have looked at the aspects that lead to an organization's adoption of BIM. BIM adoption has been boosted by a variety of factors and techniques in the past. Research on BIM adoption has focused on the relative advantages, technology, organization, environmental, human, compatibility, complexity, trialability, perceived risks, top management support, organization readiness, organizational size, and cost, which are the primary factors in an organization's ability to implement BIM and IoT as shown Figure 8. The elements that influence BIM adoption are mainly people, processes, technology, strategic IT planning, and collaborative process. BIM adoption is driven by a combination of technological, organizational, and environmental factors. Technology-Organization-Environment (TOE) framework provides the basis for this aspect. TOE is commonly seen as the most important factor influencing a company's

decision to embrace new technology. As shown in Figure 5, compatibility which was considered in 11 articles is the most widely investigated factor in BIM and IoT. Besides, factors such as relative advantage, complexity, and top management support were also widely investigated as indicated by the total number of articles. However, human factors, normative pressure, and communication beviour were less investigated. Several authors have reported the various factors that influence BIM and IoT adoption. Ahmed [73] identified 11 critical factors that influence BIM adoption and application within the UK architecture sector. The factors, willingness to adopt BIM, communication behaviour of an organization, observability of BIM benefits, compatibility of BIM, social motivations among organization's members, the relative advantage of BIM, organizational culture, top management support, organizational readiness, coercive pressures, and organization size. According to the findings, the relative advantage of BIM is the aspect that is most influential and crucial across all three stages of the adoption process namely, the awareness stage, the intention stage, and the decision stage of the BIM adoption process. Using the Chinese construction industry as an example, Chin and Yin [74] developed a study model that incorporates the critical factors that linked BIM technology, the construction firm, and the environment. According to the findings, a key factor that facilitated BIM acceptance was the relative advantage, whereas a factor that inhibited BIM adoption was the complexity. In addition, top management support was a crucial factor that contributed to the adoption of BIM. However, organizational preparedness was significant for engineering consulting businesses but not for construction firms. Ezeokoli et al. [75] identified 12 factors that influence BIM adaptability in construction projects in Nigeria. The study revealed that most of the BIM potential is not being utilized because of a variety of factors such as incompatibility between different software platforms, a lack of industry knowledge and awareness, industry structure and culture, a lack of appropriate technology and infrastructure, implementation costs, and individual/personal disposition. McCartney [76]. Liao et al. [77] identified 21 critical factors that hinder BIM implementation in building projects in Singapore. Among the various factors identified, lack of executive vision and training was recommended to have top management priority

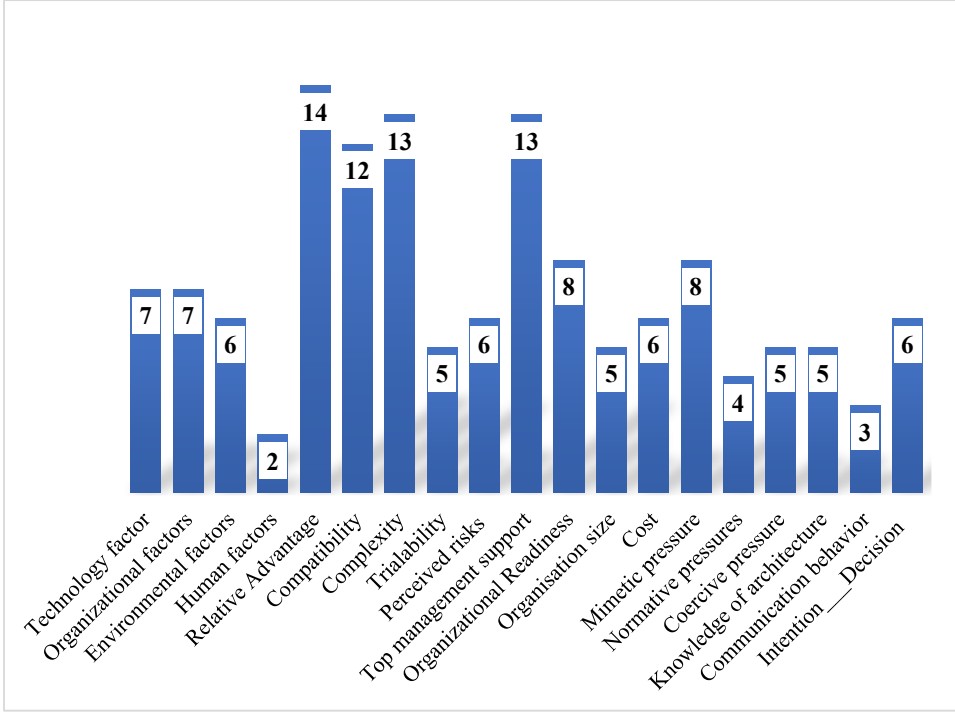

**Figure 8.** Number of articles with the various extracted factors.

## 4. Conclusions

The concept of combining building information modelling (BIM) with data sources coming from the Internet of Things (IoT) is a new one. BIM and IoT data, for the most part, offer complementary perspectives on the project that balance each other's limits. The analysis of previous studies reported in this study revealed that using the BIM-IoT idea in the construction sector, which has been identified as having a high-risk component, could improve overall performance while lowering the risks associated with operations and procedures. This fulfilled the goal of this study which is to establish the feasibility of integrating BIM-IoTs in the construction industry and give recommendations based on the available data. BIM and IoT have been widely employed in construction projects for several purposes, including construction safety risk assessment, dispute management, building construction sustainability, and on-site construction process monitoring according to literature trends examined. On the other hand, there is a lack of research awareness of the prospects of BIM-IoT integration in the construction industry which could become a hot topic in the quest for an innovative construction industry.

**Author Contributions:** Conceptualization, methodology, software, formal analysis, writing original draft preparation, project administration, B.H.M.; review, and editing, visualization, supervision, funding acquisition, H.S., E.Y., N.S.M.S., A.H.B.H. and S.A. All authors have read and agreed to the published version of the manuscript.

**Funding:** This work was supported in part by Universiti Kebangsaan Malaysia, and in part by the Ministry of Higher Education Malaysia under Grant (FRGS/1/2020/TK0/UKM/02/9 and FRGS/1/2021/TK01/UKM/02/1).

**Institutional Review Board Statement:** Not applicable.

**Informed Consent Statement:** Not applicable.

**Data Availability Statement:** Not applicable.

**Conflicts of Interest:** The authors declare no conflict of interest.

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
