# Peer review of "Nexus between Building Information Modeling and Internet of Things in the Construction Industries"

_applsci, doi:10.3390/app122010629_

Round 1

Reviewer 1 Report

Refer attachment

Author Response

Reviewer 1

 We thank Reviewer 1 for critically reviewing our manuscript. We have provided responses and rebuttals to the comments raised. The changes made are highlighted in yellow throughout the manuscript.

General Comments:

  1. Introduction is well explained with the latest articles cited from Scopus and Science Direct to make readers understand the main concept of BIM and IoT.

Response

Thank you for the compliments.

  1. However, the authors did not mention problems before BIM and IoT are implemented. It is very nice if they are able to describe, what are problems in the current situation in the construction industry. Therefore, they came out with BIM and IoT integration.

Response

Thank you for the comment. We have included the problem statement prior to discussing the integration of BIM and IoT in the revised manuscript as indicated in yellow highlights. Please, refer to lines 373-376.

  1. The papers also provided illustrations and figures to better understand the process of the analysis done, but some figures are missing. Please check the figures and explanations carefully.

Response

Thank you for the observation. We have revised the manuscript and properly placed the missing Figures.

Specific Comments:

  1. In the abstract, the finding of the study is missing.

Response

Thank you for the response. We have included some of the key findings from the study in the abstract of the revised manuscript. Please, refer to lines 21-25.

  1. Specifically, this paper is well-explained relevant theories that could be matched with BIM and IoT applications.

Response

Thank you for the compliments

Reviewer 2 Report

Dear authors,

Thanks for your contribution to Applied Sciences.

Before further process of this manuscript, please check if it matches the scope of the journal.

With major revision of the manuscript, it might be accepted.

The opinions are set out below:

Structure

Please prepare the manuscript following the instructions for authors. 

English

The manuscript has several typos. Authors need to proofread the paper to eliminate all of them. Some sentences are too long. Generally, it is preferable to write short sentences with one idea in each sentence.

References

The literature review is incomplete. Several relevant references are missing. The reference list should include the full title, as recommended by the style guide.

Introduction 

Authors should include additional references in the introduction that support the claims. Authors should better explain the background to this research, including why the research issue is important. Contributions should be enhanced. It should be made clear what is novel and how it addresses the limitations of prior work.

Related work 

The related work section is not well organized. Writers should try to categorize articles and present them logically. Authors should add a table comparing the main features of previous work in order to highlight their differences and limitations. Alternatively, authors may consider adding a row to the table to describe the proposed solution.

Problem definition 

Authors should provide a clear and detailed definition of the issue. Authors should include an example to illustrate how the problem is defined.

Method

A novel solution is presented, but it is important to better explain the design decisions (e.g. why the solution is designed that way). There is a need for discussion of the complexity of the proposed solution. A PRISMA method is highly recommended to proceed the review articles.

Case study

Case study should be updated to incorporate some comparisons with more recent studies.

Sincerely yours,

Author Response

Reviewer 2

We thank Reviewer 1 for critically reviewing our manuscript. We have provided responses and rebuttals to the comments raised. The changes made are highlighted in yellow throughout the manuscript.

English

The manuscript has several typos. Authors need to proofread the paper to eliminate all of them. Some sentences are too long. Generally, it is preferable to write short sentences with one idea in each sentence.

Response

Thank you for the observation. We have proofread the manuscript and corrected the typo errors as indicated in yellow highlights. The sentences have also been revised.

References

The literature review is incomplete. Several relevant references are missing. The reference list should include the full title, as recommended by the style guide.

Response

We have revised the reference list and updated it accordingly. They have been formatted based on the Journal’s guideline.

Introduction

Authors should include additional references in the introduction that support the claims. Authors should better explain the background to this research, including why the research issue is important. Contributions should be enhanced. It should be made clear what is novel and how it addresses the limitations of prior work.

Response

Thank you for the comment. We have included additional references in the revised introduction. The background of the research has been improved. The contribution and novelty have been highlighted in the last paragraph of the revised manuscript.

Related work

The related work section is not well organized. Writers should try to categorize articles and present them logically. Authors should add a table comparing the main features of previous work in order to highlight their differences and limitations. Alternatively, authors may consider adding a row to the table to describe the proposed solution.

Response

Thank you for the comment. We believe Table 1 and 2 served this purpose.

Problem definition

Authors should provide a clear and detailed definition of the issue. Authors should include an example to illustrate how the problem is defined.

Response

Thank you for the comment. The manuscript has been revised with detailed definition of issues.

Method

A novel solution is presented, but it is important to better explain the design decisions (e.g. why the solution is designed that way). There is a need for discussion of the complexity of the proposed solution. A PRISMA method is highly recommended to proceed the review articles.

Response

Thank you for the comment. The method section has been revised for clarity.

Case study

Case study should be updated to incorporate some comparisons with more recent studies.

Response

Thank you for the comment. We meant to write “Example” not “Case study” We did not employ a case study approach.

Sincerely yours,

Reviewer 3 Report

You reviewed the contents of research papers with the keywords of BIM and IOT and derived correlations. Currently, several papers are published in the MDPI journal on the topic of BIM and IoT integration.

Therefore, if you add the contents analyzed by organizing the papers published on the topic of integrating BIM and IOT, it will be quite good content.

Author Response

We thank Reviewer 3 for critically reviewing our manuscript.

Comment

You reviewed the contents of research papers with the keywords of BIM and IOT and derived correlations. Currently, several papers are published in the MDPI journal on the topic of BIM and IoT integration.

Therefore, if you add the contents analyzed by organizing the papers published on the topic of integrating BIM and IOT, it will be quite good content.

Response

Thank you for the compliments.

Reviewer 4 Report

The paper's intention is good; it tries to identify the nexus between BIM and IoT.  However, overall, it lacks critical evaluation of the materials, and the lack of argumentation is also disappointing.  It looks more like a literature report rather than a literature review. 

The authors present a few studies that have looked into the integration of BIM and IOT, so what are the trends, debates and gap(s) that motivate this study?  There are many research questions presented.  How are these questions associated with the vital need to get awareness?  Its novelty is unclear.  

The methodology part presents how initiating the search was conducted. Still, it does not explain how the storing and organising of information was done, as well as analysing and synthesising them.  How many published works have been screened? How many have met the inclusion/exclusion criteria?  Findings should be presented in its own section, not under the Methodology section.  Discussion on findings is very hard to follow, this should be improved by organising or categorising the literature chronologically, thematically or methodologically.  

This paper needs a total revision in terms of restructuring, academic argumentation, and strong evidence of analytical thinking shown through the connections made between the reviewed literature.

Author Response

Comment

The paper's intention is good; it tries to identify the nexus between BIM and IoT.  However, overall, it lacks critical evaluation of the materials, and the lack of argumentation is also disappointing.  It looks more like a literature report rather than a literature review.

We thank Reviewer 4 for critically reviewing our manuscript. We have revised the manuscript based on the comments raised. Rebuttals have been provided where it is necessary. The changes made are highlighted in yellow throughout the manuscript.

 Comment

The authors present a few studies that have looked into the integration of BIM and IOT, so what are the trends, debates and gap(s) that motivate this study?  There are many research questions presented.  How are these questions associated with the vital need to get awareness?  Its novelty is unclear. 

Response

Thank you for the comment. We have revised the manuscript to address all the trends, debate, gaps that motivate the study. The novelty of the study has been clarified in the revised manuscript.

Comment

The methodology part presents how initiating the search was conducted. Still, it does not explain how the storing and organising of information was done, as well as analysing and synthesising them.  How many published works have been screened? How many have met the inclusion/exclusion criteria?  Findings should be presented in its own section, not under the Methodology section.  Discussion on findings is very hard to follow, this should be improved by organising or categorising the literature chronologically, thematically or methodologically. 

Response

Thank you for the comment. We have separated the Results and Discussion from the methodology. We have also improved on the discussion of the findings.

Comment

This paper needs a total revision in terms of restructuring, academic argumentation, and strong evidence of analytical thinking shown through the connections made between the reviewed literature.

Response

Thank you for the comments. We have revised the manuscript accordingly.

Reviewer 5 Report

This paper is a review paper, which summarizes the existing research on Building Information Modeling (BIM), Internet of things (IoT), and their cross-disciplinary topics. Detailed comments are listed below:

Major problems:

1.      What is the main contribution of this paper? By reading it, what is expected to be learned by the readers? The main point of this paper seems to just count the number of papers using keywords, publication locations, and topics.

2.      Even as a review paper, the general methodology should be summarized to give a reference for future research. So can any commonalities of the literature be summarized as a workflow?

3.      What is the conclusion of this paper? Does it reveal the future tendency of research topics? Does it provide any useful criteria to evaluate the value of related research papers?

Minor problems:

1.      On page 1 line 45, the word “best” is too subjective. Please use the academic expression.

2.      There is a layout error in Figure 1. Also, Figure 1 does not provide new and useful information related to the topic.

3.      Please proofread the article. There are some grammar problems.

4.      Please re-arrange Figure 2 and Figure 3. The images in one figure should be displayed on the same page.

5.      The figures starting from page 14 are incorrectly indexed.

In conclusion, a major revision of the structure of this paper is required.

Author Response

We thank the Reviewer for critically reviewing our manuscript. We have provided responses and rebuttals to the comments raised. The changes made are highlighted in yellow throughout the manuscript.

Major problems:

  1. What is the main contribution of this paper? By reading it, what is expected to be learned by the readers? The main point of this paper seems to just count the number of papers using keywords, publication locations, and topics.

Response

Thank you for the comment. The main contribution of the manuscript has been highlighted in the last paragraph of the revised manuscript.

  1. Even as a review paper, the general methodology should be summarized to give a reference for future research. So can any commonalities of the literature be summarized as a workflow?

Response

Thank you for the comment. We have improved the methodology section in the revised manuscript.

  1. What is the conclusion of this paper? Does it reveal the future tendency of research topics? Does it provide any useful criteria to evaluate the value of related research papers?

Response

Thank you for the comment. We have improved the conclusion section in the revised manuscript with the addition of future tendency for research.

Minor problems:

  1. On page 1 line 45, the word “best” is too subjective. Please use the academic expression.

Response

Thank you for the comment. We have changed the word “best” to “appropriate” in the revised manuscript.

  1. There is a layout error in Figure 1. Also, Figure 1 does not provide new and useful information related to the topic.

Response

Thank you for the comment. Figure 1 depicts the diagrammatical representation of the steps involved in the methodology which we deem it fits to give an overview of the approach employed.

  1. Please proofread the article. There are some grammar problems.

Response

Thank you for the observation. The manuscript has been proofread. The corrections made are shown in yellow highlights throughout the revised manuscript.

  1. Please re-arrange Figure 2 and Figure 3. The images in one figure should be displayed on the same page.

Response

Thank you for the observation. We have rearranged the Figures in the revised manuscript as suggested.

  1. The figures starting from page 14 are incorrectly indexed.

Response

Thank you for the observation. The figures indexing has been corrected in the revised manuscript.

In conclusion, a major revision of the structure of this paper is required.

Thank you for your efforts in reviewing our manuscript.

Reviewer 6 Report

The authors examined the potential of integrating BIm-IoTs in the construction industries by examining the literature. The goals and objectives of the study are clearly explained. Thus, I will suggest only a few changes to improve paper quality.

1. Figure 1 should be developed. It can be expressed with a simpler drawing or etc.

2. In Figure 2, which keywords were used in the literature review? Please explain this before the figure. Also please use a 2D graph for Figure 2-b.

3. It is difficult to read Figure 3. Also, please change the figure name to 'Distribution of BIM and IoT studies according to countries' or etc.

4. Results and discussions should be separate chapters. The authors gave this part in the Methodology section. The results cannot be given in the methodology section. 

5.  The conclusion part should be enhanced. 

6. You should explain the implications of your study for future work in the discussion or conclusion chapter.

Author Response

We thank Reviewer 6 for critically reviewing our manuscript. We have revised the manuscript based on the comments raised. Rebuttals have been provided where they are necessary. The changes made are highlighted in yellow throughout the manuscript.

Comment

The authors examined the potential of integrating BIm-IoTs in the construction industries by examining the literature. The goals and objectives of the study are clearly explained. Thus, I will suggest only a few changes to improve paper quality.

Response

Thank you for the compliment. We have revised the manuscript based on the comments raised.

Comments

  1. Figure 1 should be developed. It can be expressed with a simpler drawing or etc.

Response

Thank you for the comment. Figure 1 has been revised.

Comment

  1. In Figure 2, which keywords were used in the literature review? Please explain this before the figure. Also please use a 2D graph for Figure 2-b.

Response

Thank you for the comment. We have specified the keywords used in the literature review. Figure 2-b has been changed to 2-D format.

Comment

  1. It is difficult to read Figure 3. Also, please change the figure name to 'Distribution of BIM and IoT studies according to countries' or etc.

Response

Thank you for the comment. We have changed Figure to 2-D. Also the Figure name  has been revised to “Distribution of BIM and IoT studies according to countries.

Comment

  1. Results and discussions should be separate chapters. The authors gave this part in the Methodology section. The results cannot be given in the methodology section.

Response

Thank you for the comment. We have separated the Results and discussion in the revised manuscript.

Comment

  1. The conclusion part should be enhanced.

Response

Thank you for the comment. The conclusion has been enhanced in the revised manuscript.

Comment

  1. You should explain the implications of your study for future work in the discussion or conclusion chapter.

Thank you for the comment. The implication of the study for future work has been explained in the conclusion of the revised manuscript.

Round 2

Reviewer 2 Report

Accept

Reviewer 4 Report

Generally, the manuscript is still read as a literature report rather than a literature review.  Authors have mentioned that rebuttals have been provided where necessary, but the yellow highlighted additions show grammatical/spelling corrections, not scientific arguments.  Because of that, this paper remains lacking in academic argumentation and strong evidence of analytical thinking.  

An improvement to explain the novelty of the research is absent in the revised manuscript.  It remains unclear. 

The methodology section has no improvement, as suggested.

Reviewer 5 Report

The authors have slightly revised the manuscript, but still haven’t answered my questions and fulfilled the requirements. I’ll explain the major problems again:

1.      What is the main contribution of this paper? By reading it, what is expected to be learned by the readers? The main point of this paper seems to just count the number of papers using keywords, publication locations, and topics.

A good review paper should provide a summarization that is helpful for future researchers to follow and define their research domains. So is the current categorization valuable, and why? What is the feature of each category and how can we tell them apart? What is the key research methodology of each category and how can we better achieve it in the future? Will new Interdisciplinary research domains appear, what is the tendency? Those are useful information for the readers.

2.      Even as a review paper, the general methodology should be summarized to give a reference for future research. So can any commonalities of the literature be summarized as a workflow?

Here, what I mean by the “general methodology” is a workflow to summarize the workflows of all categories in this article. For example, some research starts from the same objectives, but uses different techniques, and evaluates the results with different measures. So what are the commonalities and differences? How does it lead to different branches in the workflow?

3.      What is the conclusion of this paper? Does it reveal the future tendency of research topics? Does it provide any useful criteria to evaluate the value of related research papers?

Besides what I mentioned in question 1, another useful piece of information for the readers is the general method to distinguish between successful and failed research. For example, what kinds of research are more valuable and useful in practice? If the readers want to step in and do some research in this field of “BIM+IoT”, what are your suggestions? This should be reflected as a part of the conclusion or at the end of the introduction of each category.

Reviewer 6 Report

The authors have addressed the necessary adjustments. Accept in present form.